# Bidimensional lamellar assembly by coordination of peptidic homopolymers to platinum nanoparticles

Ghada Manai[1,2,8], Hend Houimel[1,8], Mathilde Rigoulet[1], Angélique Gillet[1], Pier-Francesco Fazzini [1], Alfonso Ibarra[3], Stéphanie Balor[4], Pierre Roblin[5], Jérôme Esvan[6], Yannick Coppel [2], Bruno Chaudret[1], Colin Bonduelle [2,7✉] & Simon Tricard [1✉]

A key challenge for designing hybrid materials is the development of chemical tools to control the organization of inorganic nanoobjects at low scales, from mesoscopic (~μm) to nanometric (~nm). So far, the most efficient strategy to align assemblies of nanoparticles consists in a bottom-up approach by decorating block copolymer lamellae with nanoobjects. This well accomplished procedure is nonetheless limited by the thermodynamic constraints that govern copolymer assembly, the entropy of mixing as described by the Flory–Huggins solution theory supplemented by the critical influence of the volume fraction of the block components. Here we show that a completely different approach can lead to tunable 2D lamellar organization of nanoparticles with homopolymers only, on condition that few elementary rules are respected: 1) the polymer spontaneously allows a structural preorganization, 2) the polymer owns functional groups that interact with the nanoparticle surface, 3) the nanoparticles show a surface accessible for coordination.

[1] Laboratoire de Physique et Chimie des Nano-Objets, INSA, CNRS, Université de Toulouse, Toulouse, France. [2] Laboratoire de Chimie de Coordination, CNRS, Université de Toulouse, Toulouse, France. [3] Instituto de Nanociencia de Aragón, Universidad de Zaragoza, Zaragoza, Spain. [4] Plateforme de Microscopie Électronique Intégrative, Centre de Biologie Intégrative, CNRS, Université de Toulouse, Toulouse, France. [5] Laboratoire de Génie Chimique, Fédération Fermat, INPT, CNRS, Université de Toulouse, Toulouse, France. [6] Institut Carnot – Centre Inter-universitaire de Recherche et d'Ingénierie des Matériaux, INP-ENSIACET, CNRS, Université de Toulouse, Toulouse, France. [7] Laboratoire de Chimie des Polymères Organiques, Université de Bordeaux, CNRS, Bordeaux INP, Pessac, France. [8] These authors contributed equally: Ghada Manai, Hend Houimel. ✉email: colin.bonduelle@enscbp.fr; tricard@insa-toulouse.fr

Structuring hybrid materials combining metallic nanoparticles (NPs) and polymers has stimulated a large effort to make new physical properties emerging: optical, electronic, or magnetic[1–3]. Such composites have the potential of improving the functionality of devices ranging from memory storage to sensors or microelectronic systems[4,5]. In addition, the structural properties of the polymer matrix can give added value to the hybrid materials, such as stimuli-responsiveness[6] or chirality[7]. Among a variety of possible structuring, disposing NPs in lines on substrates paves a way toward anisotropic ordering of matter at the nanoscale, which can be observed, manipulated, and connected. To reach such a supraparticular organization, the most efficient strategy developed so far consists in decorating diblock copolymer lamellar assemblies by NPs, tuning the strength and the nature of weak interactions between the two components (essentially Van de Waals interactions)[5–11]. Diblock copolymer templating is a robust and versatile approach that has been extended to thermal evaporation[12], atomic layer deposition[13], or for structuring molecules such as polyoxometallates[14]. The obtained morphologies are driven by (1) the entropy of mixing of the two blocks, described by the Flory–Huggins solution theory[15,16], and (2) their corresponding volume fractions[17], taking into account that, in presence of NPs, there is a significant decrease in the copolymers' conformational entropy coming from particle sequestration[10]. Although very promising in many situations, the diblock copolymer approach shows important limitations: (1) a constrained size ratio between the two blocks to access lamellar assemblies (generally comprised between 0.4 and 0.6 volume fraction with random coil polymers), (2) in presence of NPs, an imprecise control of their localization (as it is governed by entropy) and (3) in solution, a strong dependence to experimental conditions (concentration, temperature, presence of co-solvent, etc.)[10,15,16]. Here, we present an alternative strategy to dispose NPs in lines by simply mixing metallic ultra-small NPs with structured peptidic homopolymers. The coordination bonding between the polymer functional groups and the NP surface indeed affords lamellar organization, where the patterning distances are linearly controlled by the molecular weight of the polymer.

Self-assembly is a widely applied strategy for preparing various materials based on metallic NPs, which lead to e.g., innovative photonic materials, new microelectronic devices, or structured templates for nanolithography[1]. NPs are generally obtained by a chemical approach that consists in constraining the growth of the crystal by the presence of a limiting agent (ligand, polymer, etc.). An elegant approach resides in decomposing organometallic precursors in mild conditions to obtain clean NPs, i.e. having a surface composition perfectly described and controllable. In this work, we choose, as a standard, ultra-small (<2 nm) platinum NPs, as their sizes are in the same range as the one of a monomer moiety and their surfaces are only stabilized by carbon monoxide (CO) and labile tetrahydrofuran (THF)[18]. The absence of strong organic ligands gives the opportunity to fully describe the coordination between the additional component, i.e. the polypeptide polymer, and the NP surface. So far, most approaches involving NPs consider their assemblies as a packing of hard spheres, but the ligand itself has emerged as a chemical leverage, which enables exquisite control to promote supraparticular chemistry[19,20]. Polypeptide polymers are simple macromolecules in which an amino-acid moiety is repeated many times. They adopt ordered secondary conformations such as α-helices or β-sheets, which offer a way to guide structuration at the nanoscale through intermolecular and/or intramolecular interactions[21,22]. In this work, we choose poly(γ-benzyl-L-glutamate) (PBLG), a synthetic polypeptide that gives rise to a rigid rod-like α-helical conformation in organic solvents and that has often been employed as a model system to drive lamellar morphologies, when included in block copolymer structures[23].

## Results

**Description of the nanostructuration.** The NPs were synthesized by decomposition of $Pt_2(dba)_3$ (dba = dibenzylideneacetone) under a CO atmosphere in THF, followed by complete elimination of the organic dba residue by washing with pentane[18]. TEM pictures showed well-dispersed NPs, with diameters of $1.2 \pm 0.3$ nm (Fig. 1a). On another hand, PBLG was synthesized by initiating the ring-opening polymerization of γ-benzyl-L-glutamate-N-carboxyanhydride with propargylamine in dimethylformamide (Supplementary Fig. 1)[24]. A library of PBLGs spanning a wide range of molecular weights were obtained (Fig. 1b), with five degrees of polymerization (Dp), as measured by [1]H NMR and SEC chromatography (Supplementary Table 1): **PBLG1** presented a Dp of 28, **PBLG2** of 69, **PBLG3** of 120, **PBLG 4** of 217, and **PBLG 5** of 481. Their α-helix secondary structures were confirmed by circular dichroism measurements in THF (Fig. 1c and

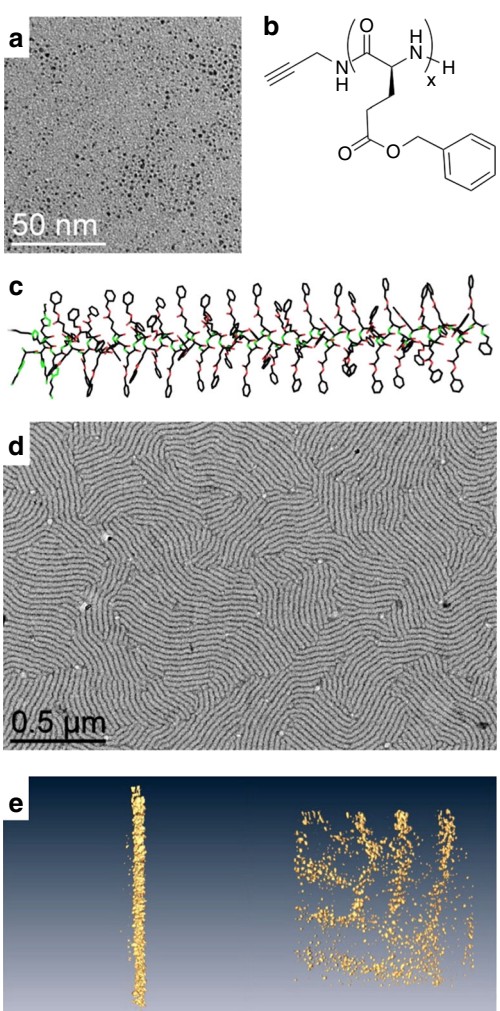

**Fig. 1 Building blocks and self-assembly. a** TEM image of pristine ultra-small platinum nanoparticles of 1.2 ± 0.3 nm. **b** Chemical structure of PBLG —in the present study, x = 28 (**PBLG1**), 69 (**PBLG2**), 120 (**PBLG3**), 217 (**PBLG4**), 481 (**PBLG5**). **c** Geometrical model of the α-helix conformation of PBLG (example for x = 60). Lamellar structuration of an assembly of platinum nanoparticles with **PBLG4** at 0.5 eq.: **d** Low-magnification TEM image; **e**, Tomography 3D reconstruction at two viewing directions: each yellow dot corresponds to an individual nanoparticle.

Supplementary Fig. 2). The assembly of NPs and peptidic polymers was carried out in THF: solutions of platinum NPs and of PBLG were mixed and stirred for 2 h at different equivalent numbers (eq. – defined as the ratio between the monomer unit and the platinum atom quantities). Transmission electron microscopy (TEM) on **PBLG4** at 0.5 eq. showed unexpected lamellar assemblies of the hybrid materials (Fig. 1d), alternating zones containing (dark), or excluding (white) NPs. Such an organization has been observed on both silicon substrates and TEM grids by different microscopy techniques: atomic force microscopy and scanning electron microscopy, in addition to TEM (Supplementary Fig. 3a–c). Tomography imaging showed that the NPs constitute cylindrical lamellae, without any preferential arrangement within each lamella (Fig. 1e and Supplementary Fig. 3d, e). As we did not notice any effect of deposition substrate (hydrophobic carbon vs. hydrophilic silicon), nor of deposition process (drop-casting vs. spin-coating), we hypothesized that these structured lamellae were present in solution and did not form during the solvent evaporation. The existence of the structured lamellae in solution was then confirmed by cryo-TEM imaging, as a same morphology was observed after a fast freezing of the assembly solution (Supplementary Fig. 4). Besides, in some cases, we noticed the presence Moiré patterns, resulting from the superimposition of up to four lamellae (Supplementary Fig. 5). As the lamellae structures are preformed in solution, they can deposit on top of each other during the drop casting of the TEM

grid preparation. Such observations gave insight that the system organized in a lamellae-within-lamellae hierarchical assembly[25]. Information on the polymer structure before and after assembly was first given by $^{13}C$ HR-MAS NMR at slow speed (Supplementary Fig. 6). After NP addition, an increase of the spinning sideband intensities, due to stronger chemical shift anisotropies, highlighted a stiffening of the PBLG in the assembly, which also led to greater conformational homogeneity, as evidenced by the resonance sharpening. Second, SAXS measurements showed a broad peak centered at 0.25 Å$^{-1}$ (Supplementary Fig. 7), meaning that NPs were separated from each other by an average correlation distance equal to 2.5 nm. The importance of the ratio between the monomer unit and the platinum atom quantities was then studied by varying the eq. number from 0.05 up to 5 eq. with **PBLG4**. Without any polymer, the NPs simply aggregated because of the capillary forces generated by THF evaporation (Fig. 2a), without showing any specific average distance by SAXS analysis (Supplementary Fig. 7). As soon as PBLG was added to the reaction mixture, depletion zones without NPs formed (Fig. 2b–d), attributed to the presence of the polymer. Very clean patterns were obtained at 0.5 eq, with alternation of regular lamellae. At higher ratios, this NP organization became looser (1 eq.—Fig. 2d), and totally disappeared at 5 eq. (Supplementary Fig. 8). A window of two orders of magnitude in eq. number (from 0.05 to 5 eq.) was thus accessible to tune the pattern structure of the lamellar assemblies.

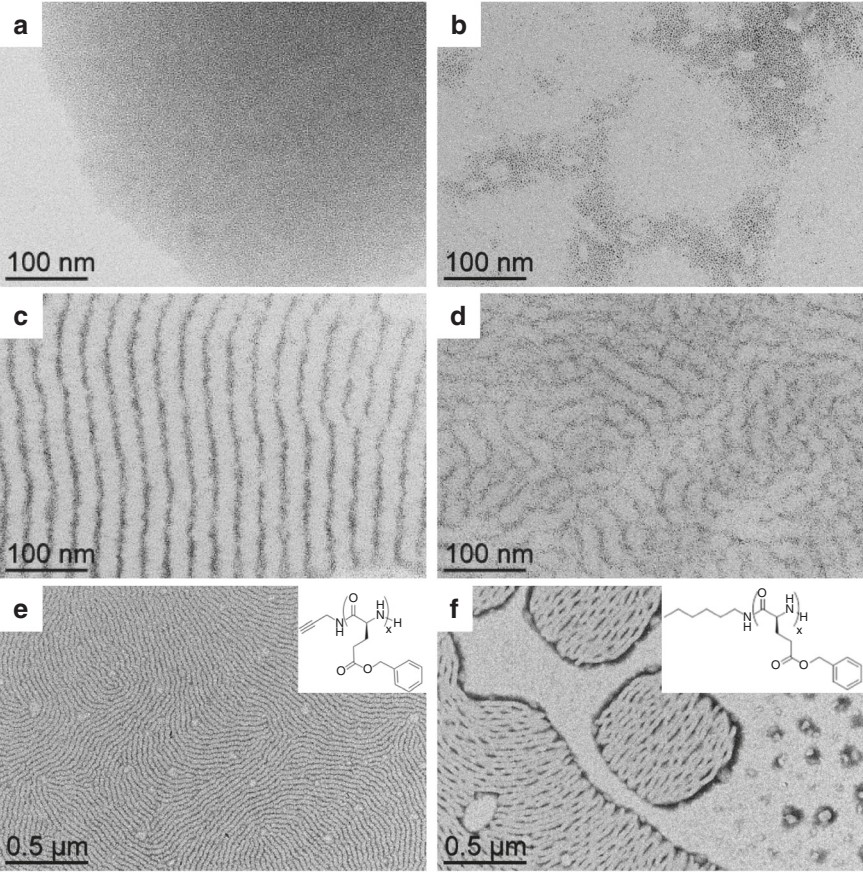

**Fig. 2 Effect of the relative quantity of polymer vs. nanoparticle and effect of the terminal group.** TEM micrographs of the assemblies after 2 h of reaction between platinum nanoparticles and: **PBLG4** at **a** 0 eq. (nanoparticles alone); **b** 0.05 eq.; **c** 0.5 eq.; **d** 1 eq. (an equivalent eq. refers to the number of introduced monomers per platinum atom). The assembly process occurs at an optimum relative ratio of polymer vs. nanoparticle equal to 0.5 eq. TEM micrograph of the assemblies between platinum nanoparticles and: **e** PBLG4; **f** PBLG-H, at 0.5 eq., after 2 h of reaction. Insets represent the chemical structures of the polymers. The nature of the terminal group of PBLG influences the global cohesion of the hybrid system, as alkyne groups lead to better structuration than hexyl groups.

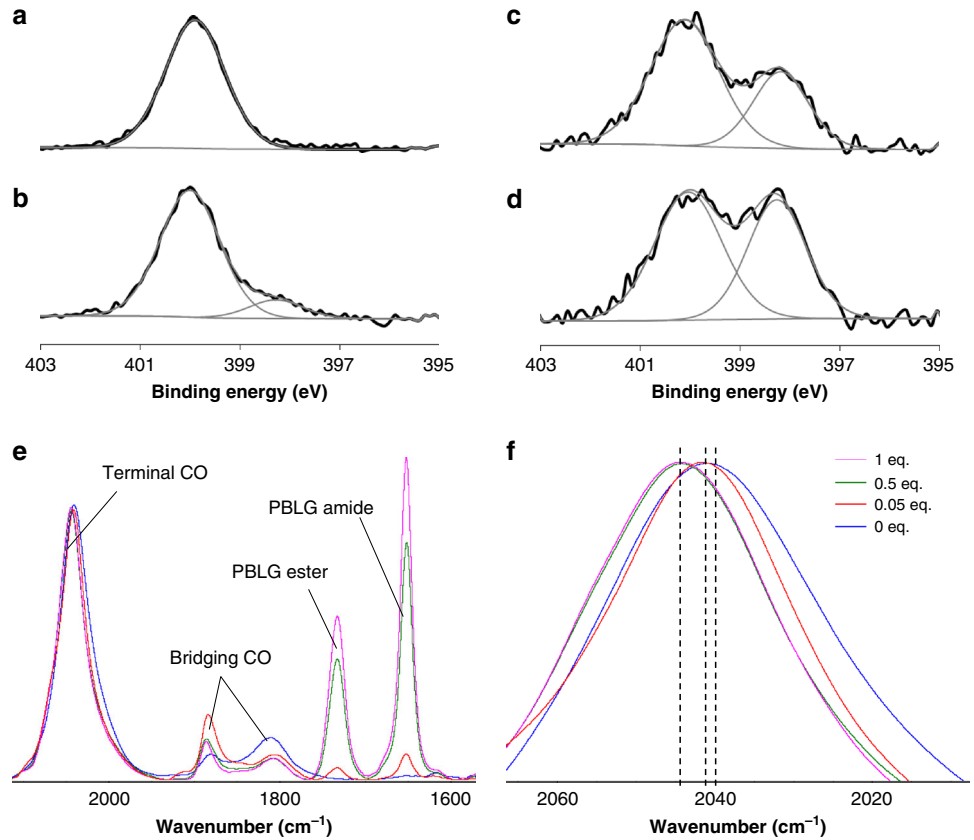

**Fig. 3 Spectroscopic signature of coordination of the peptidic polymer to the nanoparticle surface. a** XPS spectrum at the N1s edge of **PBLG4** alone. XPS spectra at the N1s edge of assemblies between platinum nanoparticles and **PBLG4** at: **b** 1 eq. (12% of component at 398 eV); **c** 0.5 eq. (33% of component at 398 eV); **d** 0.05 eq. (45% of component at 398 eV). **e** Infrared spectra of CO coordinated at the nanoparticle surface and of peptide bond of the polymer within the assemblies (at 0 eq., 0.05 eq., 0.5 eq., and 1 eq.). The spectra are normalized to the signal of the CO vibration around 2040 cm$^{-1}$. **f** Zoom on the infrared spectra of Fig. 2e at the terminal CO region. The assembly process is characterized by a specific signature both in XPS and infrared spectroscopies.

**Origin of the patterning**. To determine key parameters at the origin of the patterning, coordination of the polymer to the NP surface was studied. First, a peptidic dimer **DBLG** was specifically prepared and mixed with NPs to confirm the possible coordination of the alkyne moiety at the NP surface. $^{13}$C solid-state MAS NMR showed the disappearance of the peaks at 72 and 80 ppm attributable to the alkyne moiety upon NP addition, whereas the other peaks of the molecule remained unchanged (Supplementary Fig. 9). This disappearance strongly supported the occurrence of similar alkyne coordination to the NP surface with **PBLG4**. A polymeric analogue without alkyne was then synthesized from hexylamine, **PBLG-H**. After mixing with NPs, TEM imaging showed a significantly less structured system, with free NPs and more discontinuous NP arrangement within the lamellae (Fig. 2e, f), thus confirming the importance of the alkyne coordination. Second, another coordination bonding between the NP surface and the polymer was identified by X-ray photoelectron spectroscopy (XPS) at the N1s edge, which showed the appearance of a new peak at 398 eV in addition to the neutral component at 400 eV of the free polymer. The relative intensity of this peak significantly increased when the eq. number decreased (Fig. 3a–d and Supplementary Fig. 10), confirming a rise of the electronic density on the PBLG amide nitrogen upon NP addition. Simultaneously, both the relative increase of surface platinum oxidation measured by XPS at the Pt4f edge (Supplementary Fig. 11), and the progressive shift of the adsorbed CO peak in infrared spectroscopy from 2040 cm$^{-1}$ at 0 eq. to 2041 cm$^{-1}$ at 0.05 eq. and to 2045 cm$^{-1}$ at 0.5 and 1 eq. confirmed a decrease of the electronic density at the NP surface upon PBLG addition

(Fig. 3e, f and Supplementary Fig. 12)[18]. Overall, both XPS and infrared measurements reflected an electronic transfer from the NP surface to reduce the PBLG nitrogen in the composite materials, as already observed for amides in platinum molecular complexes[26,27]. The present study thus indicates that the polymer interacts with the NP surface by both coordinating the terminal alkyne group of the polypeptide and its peptide linkages.

**Influence of the degree of polymerization**. We further explored the Dp influence on the lamellar assemblies. No arrangement was observed with **PBLG1**, but image analysis showed that the width of the lamellae average periodicity regularly increased from **PBLG2** to **PBLG5** (Fig. 4, Supplementary Fig. 13, and Supplementary Table 2). Such a tendency was confirmed by SAXS measurements (Supplementary Fig. 14 and Supplementary Table 2). In addition, the average width of the white zones containing the polymer was equal at any Dp to the length of the PBLG model in α-helices conformation. Similarly, the evolution of the average width of the dark zones containing the NPs as a function of Dp ($x$) could be fitted by the diameter of gyration of a random coil model $R_g = R_0 x^\nu$. The $\nu$ value was fixed to 0.6, as predicted by theory and confirmed by experiments for excluded-volume chains[28]. Extensive study on chemically unfolded proteins found a $R_0$ value equal to 1.33[28], whereas the fit of our experimental data gave a $R_0$ value equal to 0.22, thus divided by six. Such a result can be interpreted by the presence of an average of 6–7 bridges within the polymer[29], which is coherent considering the PBLG polymer in a coil disordered state, internally connected

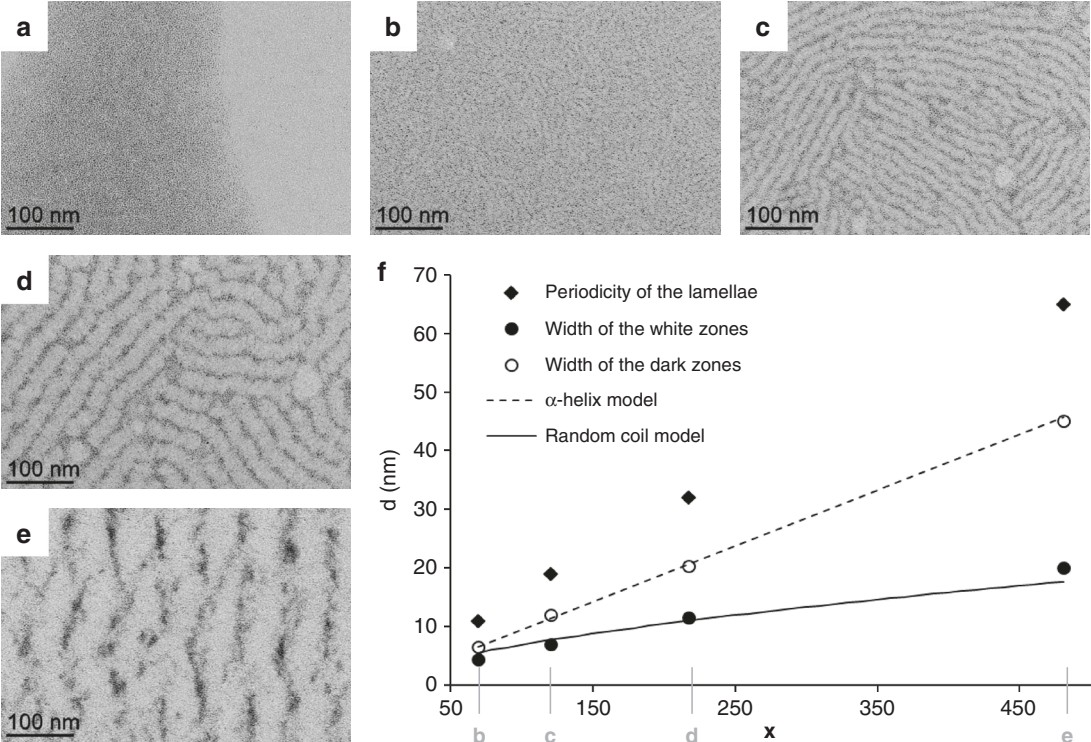

**Fig. 4 Effect of the degree of polymerization.** TEM micrographs of assemblies between platinum nanoparticles and: **a** PBLG1; **b** PBLG2; **c** PBLG3; **d** PBLG4; and **e** PBLG5. **f**, Evolution of average characteristics as a function of the degree of polymerization $x$ (the α-helix model curve represents the length of the polymer as shown in Fig. 1c for $x = 60$; the random coil model curve represents the diameter of gyration evolution of excluded volume random coil model with $R_0 = 0.22$ and $\nu = 0.6$). Positions corresponding to the micrographs of Fig. 4**b**, **c**, **d**, and **e** are made explicit on the x-axis.

by NP bridges. Analysis of tomography reconstruction confirmed that the thickness of the NP containing zones was comparable to their width (Supplementary Fig. 15). Interpretation of TEM imaging is thus in line with geometrical models where aligned polymers in α-helices alternate with alignments of coiled polymers interacting with NPs. Similar dependence with Dp was obtained if the ratio of polymer number over NP was kept constant (Supplementary Fig. 16, here the eq. number varied from 0.06 to 0.81), confirming the robustness of the patterning to variations of the polymer to NP ratio. Our assembly approach is thus a robust and simple alternative to diblock copolymer templating for structuring metallic NPs. It allows a large patterning period window comprised between 10 and 100 nm, depending on the polypeptide Dp, and can be easily described by simple geometrical models of polymers (α-helices and excluded-volume chains).

**First steps of the assembly.** In order to give insight on the lamellae formation mechanism, a time dependent study was performed with **PBLG3** and **PBLG4** (Supplementary Fig. 17). Although the standard assembly time was set to 2 h to be sure to reach a steady state (Supplementary Fig. 17a, b), we noticed that the lamellar structure was already present after five seconds, but significantly less advanced (Supplementary Fig. 17c, d). Some free NPs were indeed not assembled and aggregated around the structured zones, and the NPs lamellae were discontinuous. In order to probe effects of a fraction of second of mixing, we first deposited the PBLG on the TEM grid and we added the NP in a second time (Supplementary Fig. 17e, f). We confirmed a pre-structuration of the polymer materials, illustrated by depletion zones without NP, and a beginning of decoration of such zones by the NP, which will then interact with and coordinate to the functional groups of the peptidic polymer.

## Discussion

The results of the present study show it is possible (1) to generate anisotropic ordering of platinum NPs in lamellar assemblies by mixing with peptidic homopolymers and (2) to easily tune the dimension of these lamellar assemblies by simply varying the molar mass of the homopolymer. Generally, NP-homopolymer systems do not give extended nanostructured materials but rosaries of NPs that decorate the polymer[30]. Our results can be explained by the fact that PBLG adopt ordered secondary conformation, α-helices, which could spontaneously interact and align with each other following a nematic liquid crystal behavior[23]. The addition of NPs does not result in a simple decoration of the liquid crystal[31,32], and we hypothesized that a destructuring of some of the α-helices occurred thanks to coordination of the peptide linkages with the NP surface (a scheme of the self-assembly process is presented on Supplementary Fig. 18). This NP/polymer association would then drive a small amount of PBLG to adopt a coil-disordered state, where the NPs bridge some part of the polymers. The demixing of the two resulting phases (the NP/coil polymer hybrid on one hand and the aligned α-helices on the other hand) was facilitated by the coordination of the terminal alkyne groups on the NP surface, and led to an anisotropic ordering of NPs. In the future, to ensure optimal performances, rational design of lamellar NP assembly including polymer should consider functional and structured polymers such as peptidic polymers and a fine-tuning of the chemical interaction between the anchoring moieties of the polymers and the NP surface.

## Methods

**Starting materials.** All chemicals were purchased from Sigma-Aldrich and used as received. γ-benzyl-*L*-glutamate N-carboxyanhydride (γ-BLG NCA) was purchased from Isochem. Propargylamine and hexadecylamine were distilled before use. DMF and THF were obtained from a Solvent Purification System and freshly used.

**General procedure for the synthesis of PBLGs**. The NCA monomer of γ-benzyl-L-glutamate (BLG-NCA, 2 g, 7.6 mmol) was weighed in a glovebox under pure argon, introduced in a flame-dried schlenk, and dissolved with 4 mL of anhydrous DMF. The solution was stirred for 10 min, and propargylamine (for instance for **PBLG4**, 2 µL, 0.03 mmol) was added with an argon purged syringe. The solution was stirred for 3 days at room temperature under argon. The polymer was then recovered by precipitation in diethylether and dried under high vacuum, analyzed by $^1$H NMR (CDCl$_3$ + 15% trifluoroacetic acid). Yield: 81–92%. Molar masses were first determined by $^1$H NMR using the intensity of methylene protons of the initiator at 3.9 ppm and the intensity of methylene protons of the PBLG at 5.1 ppm. Representative $^1$H-NMR of the polypeptide backbone (400 MHz, δ, ppm): 2.13 (m, 2H,CH$_2$), 2.59 (t, 2H, CH$_2$, $J$ = 7.09 Hz), 4.37 (t, 1H, CH, $J$ = 6.56 Hz), 5.13 (s, 2H, CH$_2$O), 6.75 (s, 1H, NH), 7.35 (m, 5H, ArH)[24]. **PBLG1** presented a Dp of 25, **PBLG 2** a Dp of 59, **PBLG3** a Dp of 92, **PBLG 4** a Dp of 171, and **PBLG 5** a Dp of 373 (Supplementary Table 1). **PBLG-H** was synthesized following the same procedure but was initiated by hexylamine instead. Polymer molar masses were determined by SEC using dimethyformamide (DMF + LiBr 1 g L$^{-1}$) as the eluent. Measurements were performed on an Ultimate 3000 system from Thermoscientific equipped with diode array detector DAD. The system also includes a multi-angles light scattering detector MALS and differential refractive index detector dRI from Wyatt technology. Polymers were separated on three Shodex Asahipack gel columns [GF-1G 7B (7.5 × 8 mm), GF 310 (7.5 × 300 mm), GF510 (7.5 × 300), exclusion limits from 500–300 000 Da] at a flowrate of 0.5 mL min$^{-1}$. Easivial kit of Polystyrene from Agilent was used as a standard (Mn from 162 to 364,000 Da). Individual offline batch-mode measurements were performed to determine the homopolymers accurate refractive index increments (*dn/dc*) at 50 °C. All the samples (5 mg mL$^{-1}$) were dissolved in DMF and were run at a flow rate of 0.5 mL min$^{-1}$ at 55 °C: **PBLG1** presented a Dp of 28, **PBLG2** a Dp of 69, **PBLG3** a Dp of 120, **PBLG 4** a Dp of 217, and **PBLG 5** a Dp of 481.

The secondary structure of the PBLG blocks were studied by CD spectroscopy in THF using the following procedure: the final concentration (the concentration in the cuvette used for the CD analyses) was 180 µM in monomer units. The pathlength used was 0.01 mm to decrease the THF UV absorbance and to access correct CD signal down to 200 nm. In these conditions, the CD monitoring was performed in high resolution mode. The molar ellipticity also called the mean residue ellipticity has been calculated as follow: $[\phi] = (10\ q_{obs})/(l \times c)$. $[\phi]$ is expressed in deg cm$^2$ dmol$^{-1}$. $q_{obs}$ was the observed ellipticity in degrees (deg), $l$ is the path length in dm, and $c$ is the polypeptide concentration in mol L$^{-1}$. The range from 190 to 250 nm corresponds to the peptide bond absorption. The CD shape of all PBLGs presented two minima at about 208 and 222 nm that were attributable to α-helical structuring (Supplementary Fig. 1 for PBLG4 for which the CD signature of the helix displayed a slightly smaller 208 nm minimum as compared with a 222 nm minimum: $[\phi]$ value of −11.52 mdeg cm$^2$ dmol$^{-1}$ at 208 nm and −12.59 mdeg cm$^2$ dmol$^{-1}$ at 222 nm)[22].

**Synthesis of DBLG**. The NCA monomer of γ-benzyl-L-glutamate (BLG-NCA, 2 g, 7.6 mmol) was weighed in a glovebox under pure argon, introduced in a flame-dried schlenk, and dissolved with 6 mL of anhydrous DMF. The solution was stirred for 10 min. at 0 °C, and 1 mL of a DMF solution containing propargylamine (243 µL, 3.8 mmol) was added with an argon purged syringe. The solution was stirred for 3 h at 0 °C under argon. Upon lyophilization, the crude residue was purified by chromatography on silica gel using CH$_2$Cl$_2$/MeOH as an eluent. The dimer was isolated as a white solid (54%, 0.9 g).

**Synthesis of platinum nanoparticles**. The PtNPs have been synthesized as follows[18,33]: all operations were carried out using Fischer–Porter bottle techniques under argon. A solution of Pt$_2$(dba)$_3$ (90 mg; 0.165 mmol of Pt) in 20 mL of freshly distilled and deoxygenated THF was pressurized in a Fischer–Porter bottle with 1 bar of CO during 30 min at room temperature under vigorous stirring. During this time, the solution color changed from violet to brown (attesting the formation of the NPs). The mixture was evaporated and washed with pentane to eliminate the dba (3 × 20 mL), and to obtain native NPs. The colloid was then redissolved in 20 mL of THF. The size of the NPs was found to be equal to 1.2 ± 0.3 nm. For each series of measurements, the sizes were determined by TEM imaging.

**Self-assembly**. 1 mL of a solution of PBLG polymer in THF (2 mg in 1 mL for 0.50 eq.) was added to 4 mL of the native nanoparticle mixture under vigorous mixing. The precursor concentrations were adapted to obtain the desired equivalent of PBLG monomer per introduced Pt. The brown solution was agitated for 2 h. Drops of the crude solution were deposited on specific substrates for each characterization (see below).

**Spectroscopy for PBLG polymers**. $^1$H NMR spectra were recorded on a Bruker AC 400 spectrometer.

For circular dichroism in THF: CD spectra were recorded on a JASCO J-815 Spectropolarimeter between 205 and 260 nm (far-UV), by using a quartz cell of 0.1 cm path length, at 20 °C. The measure parameters were optimized as follows: high sensitivity, between 5 and 20 mdeg, 0.01 mdeg resolution, 8 s response time (digital integration time), 1 nm bandwidth and 5 nm min$^{-1}$ scanning rate.

**Microscopy**. Samples for TEM were prepared by deposition of one drop of the crude solution on a carbon covered holey copper grid. TEM analyses were performed at the centre de microcaractérisation Raimond Castaing using a JEOL JEM 1400 electron microscope operating at 120 kV. The mean size of the particles and the mean widths of the white and dark zones of the lamellae were determined by image analysis on a large number of objects (~300) using the ImageJ software. The auto-correlation analysis, for determining the average periodicity of the lamellae (shown on Supplementary Fig. 13), has been perform with Gatan DigitalMicrograph software. Low resolution electron tomography has been performed on a JEOL JEM 1400 microscope operated at 120 kV installed in the METI platform in Toulouse. Angles between −60° and 60° with a 2° interval where used for the acquisition. The 3D volume reconstruction has been obtained using the weighted back projection algorithm in IMOD. High resolution STEM HAADF tomography was performed at the Advanced Microscopy Laboratory (LMA), Instituto Universitario de Nanociencia de Aragon (INA), Zaragoza, Spain, by a FEI Tecnai field emission gun operated at 300 kV. 3D reconstruction was carried out with FEI tomography acquisition software, Inspect 3D and Amira 3D reconstruction software after the acquisition of 140 images.

Cryo-TEM has been performed on a JEOL 2100 microscope, equipped with a LaB6 cathode, and operated at 200 kV under low dose conditions. To prepare the samples, 3 µL of sample were deposited onto glow-discharged lacey carbon grids and placed in the thermostatic chamber of a Leica EM-GP automatic plunge freezer, set at 20 °C and low humidity. Excess solution was removed by blotting with Whatman n°1 filter paper for 0.5 s, and the grids were immediately flash frozen in liquid nitrogen. The frozen specimens were placed in a Gatan 626 cryo-holder for imaging. Images were acquired with SerialEM software, with defocus of 1–2 µm, on a Gatan US4000 CCD camera. This device was placed at the end of a GIF Quantum energy filter (Gatan, Inc.), operated in zero-energy-loss mode, with a slit width of 25 eV. Images were recorded at a nominal magnification of 4000 corresponding to calibrated pixel sizes of 1.71 Å.

AFM images were performed with an AIST-NT SmartSPM 1000 microscope. We used silicon tips (Mikromash HQNSC15/ALBS). SEM images were acquired. For both AFM and SEM experiments, the samples were prepared by drop casting of one drop of the crude solution on silicon wafers.

FT-IR spectra were recorded on a Thermo Scientific Nicolet 6700 FT-IR spectrometer in the range 4000–700 cm$^{-1}$, using a Smart Orbit ATR platform. The sample deposition was performed by drop casting of the crude solution on the germanium crystal of the platform; the measurement was acquired after evaporation of the THF solvent.

**Diffraction measurement**. X-ray diffraction patterns were recorded on a PANalytical Empyrean diffractometer using the Co Ka radiation. Small angle measurements were performed on a microscopy glass, on which the crude solution was drop-casted. An advantage of working with particles smaller than 2 nm is that the inter-particle distance is sufficiently small to observe correlation distances between two particles with a regular XRD diffractometer without the need of any dedicated SAXS facilities.

Regular Small Angle X-Ray Scattering (SAXS) measurements were performed on a XEUSS 2.0 laboratory source equipped with a pixel detector PILATUS 1 M (DECTRIS) and an X-ray source provided by GeniX3D with a fixed wavelength based on Cu Kα radiation ($\lambda$ = 1.54 Å). The sample to detector distance was fixed at 1216.5 mm giving a $q$ range starting from 0.005 to 0.5 Å$^{-1}$ assuming that $q$ is the scattering vector equal to $4\pi/\lambda$ sin $\theta$ with $2\theta$ the scattering angle. The distance was calibrated in the small angle region using silver behenate (d001 = 58.34 Å). Measurements were performed on samples in solution in capillaries. Concentration of the sample was necessary to observe a signal, so that measurements have been performed on a system that started to precipitate. The capillaries were sealed to prevent solvent evaporation and traces of water, and placed on motorized sample holder. To remove scattering and absorption from air, a primary vacuum has been applied to the entire instrument. Acquisition time per sample was set to 1 h and all scattering curves were corrected for the solvent and capillary contributions, divided by the transmission factor, acquisition time and optical path in order to obtain SAXS curves in absolute units (cm$^{-1}$).

**Spectroscopy**. X-Ray Photoelectron Spectroscopy (XPS) analyses were performed at CIRIMAT Laboratory (Toulouse) using a Thermoelectron Kalpha device. The photoelectron emission spectra were recorded using Al-Kα radiation ($h\nu$ = 1486.6 eV) from a monochromatized source. The analyzed area was about 0.15 mm$^2$. The pass energy was fixed at 40 eV. The spectrometer energy calibration was made using the C1s (284.5 ± 0.1 eV) photoelectron lines. XPS spectra were recorded in direct mode N(Ec). The background signal was removed using the Shirley method. The atomic concentrations were determined from photoelectron peak areas using the atomic sensitivity factors reported by Scofield, taking into account the transmission function of the analyzer. The photoelectron peaks were analyzed by Gaussian/Lorentzian (G/L = 50) peak fitting.

Solid-state NMR experiments were recorded on a Bruker Avance III HD 400 spectrometer equipped with a 4 mm probehead. Samples were wetted with 20 µl of THF-d$_8$ and spun between 1 and 5 kHz at 293 K. $^1$H MAS was performed with DEPTH pulse sequence and a relaxation delay of 3 s. For $^{13}$C MAS, single

pulse experiments were performed with a recycle delay of 2 s. All chemical shifts for $^{13}C$ and $^{1}H$ are relative to TMS.

## Data availability
Data are provided in the article or in Supplementary information. Original data are available from the corresponding authors upon reasonable request.

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

## Acknowledgements
We thank Abdellaziz Jouaiti and Rodrigo Fernandez-Pacheco for insightful discussions, and Vanessa Soldan for assistance for cryo-TEM imaging. Financial support from Université de Toulouse (NaSAPeP grant APR), from CNRS (MITI interdisciplinary programs, COCOPIE project) and from Agence Nationale de la Recherche (PhoCatSA grant ANR-10-LABX-0037-NEXT, and MOSC grant ANR-18-CE09-0007) is acknowledged. This study has been partially supported through the EUR grant NanoX n° ANR-17-EURE-0009 in the framework of the Programme des Investissements d'Avenir.

## Author contributions
S.T., C.B., G.M., H.H., M.R., and A.G. performed the experimental syntheses and characterizations; P-F.F. and A.I. performed tomography imaging by TEM; S.B. performed cryo-TEM imaging; P.R. performed SAXS measurements; J.E. performed XPS measurements; Y.C. performed HRMAS NMR measurements; S.T., C.B., and B.C. supervised the project; S.T. and C.B. conceived the project and wrote the manuscript. All authors discussed the results and commented on the manuscript.

## Competing interests
The authors declare no competing interests.
