## [Peer Review File · Nature Communications]

Reviewers' comments:

Reviewer #1 (Remarks to the Author):

The authors produced mixtures of nanoparticles and a homopolymer in THF as solvent. The nanoparticles were very spherical Pt nanoparticles with a diameter of 1.2 ± 0.3 nm as determined with electron microscopy. The homopolymer was poly(γ -benzyl-L-glutamate) with different molar masses. In the absence of nanoparticles this polymer forms an α helix in THF. The new finding of the work is that lamellar nanostructures are formed upon mixing. This lamellar phase consists of alternating lamellae of type 1 and type 2. The lamellae of type 1 are enriched in nanoparticles while no nanoparticles are present in type 2. This was visualized with TEM and found for a wide range of compositions.

Similar nanostructures are well reported for block-copolymers but not for homopolymers. The explanation for such nanoparticle-enriched lamellae in block-copolymers is straight-forward as can be found in the literature cited by the authors.

Normally one expects that either the polymer decorates the nanoparticle or vice versa the nanoparticles decorate the polymer. Such depend to the molar masses, concentrations, strength of interaction of the polymers and the nanoparticles etc.

But the formation of lamellar structures of the named homopolymer is highly unexpected and to me not sufficiently explained by the authors. For each polymer chain the nanoparticles need to induce a change of the conformation from an α helix to a random coil. But only for that fraction of the chain which is then located in the nanoparticle-rich lamellae type 1. The remaining part of each chain is still in an α -helical conformation forming lamellae type 2. When assuming this mechanism, a kind of pseudo block-copolymer is formed: therein block 1 is the random coil and block 2 is the α helix. In principle such is possible and can explain the lamellar structures. But such scenario is very unlikely.

Instead, I assume that the lamellar structures are formed during deposition of the mixtures on surfaces prior to TEM measurements. It needs to be excluded that a phenomenon similarly to the coffee ring effect produces the lamellar structures. Further experiments for ensemble averaging data are needed.

The authors are already going in the right direction with their SAXS measurements. But the SAXS data presented in Figure S6 is absolutely not sufficient. The peak in the scattering curve could simply result from a so-called correlation hole as found for many polymeric structures. The peak is no proper evidence for the existence of a lamellar structure in solution. Here a reasonable modelling of the SAXS curves is needed and data should be measured from all the samples that correspond to the TEM pictures presented in Figures 1 and 2. The SAXS measurements should be performed from samples in solution in a capillary. Otherwise the data are prone to artifacts. A normal lab SAXS instrument is suitable. The lamellar structures must provide a very strong scattering intensity and a distinctive scattering pattern, if they exist in solution as claimed by the authors. Data evaluation can be performed with open source small-angle scattering programs. These programs provide appropriate models for lamellar structures to simulate the measured curves.

In summary, the manuscript needs major revision that provides clear evidence for the presence of the lamellar structure in solution.

Reviewer #2 (Remarks to the Author):

The entitled work "Beyond Flory-Huggins theory: bidimensional lamellar assembly by coordination of peptidic homopolymers to platinum nanoparticles" by Manai et al. reported the use of homopolymers to control the organization of nanoparticles. The authors confirmed that PBLG polypeptide homopolymers coordinated with nanoparticles by ^{13}C NMR spectroscopy. We could imagine that the homopolymers formed brush layers on the surfaces of nanoparticles, which makes this system perfectly fit into the frame of the Flory-Huggins theory. In previous studies, the

phase behaviors of the conjugates (or amphiphiles) of nanoparticles, e.g. Au NPs (Mei et al. Chem Eur J 2019, 1) or POSS (Wu et al. Macromolecules 2014, 4622; Zhang et al. Soft Matter 2019, 7108), and polymers have been well documented. The PBLG-Pt nanoparticle conjugates in this work were in-situ generated upon the mixing of PBLG and Pt nanoparticles, which made it different from previous reports on ex-situ synthesized nanoparticle-polymer conjugates. This work does not have enough novelty for publication in Nature Communications.

Additional comments:

1. The authors claimed that the lamellar structures formed in the solution, but they characterized the films after the evaporation of solvent. It would be more convincing to use cryo-TEM to characterize the solution structure.

2. The exact structure is still unclear. It will be helpful to capture the structures at the early stage and characterize the kinetics of the self-assembly process.

3. The use of "Beyond the Flory-Huggins theory" in the title might not be appropriate.

Reviewer #3 (Remarks to the Author):

The manuscript by Manai et al. describes the self-assembly of a mixture of small platinum nanoparticles (NP) and poly(benzyl-L-glutamate) (PBLG) into periodic lamellar structures. The formation of microphase-separated structures is driven by the coordination interactions between the surface of NPs and the functional groups of PBLG. The authors systematically investigated the effect of substrates, polymer length, and volume ratio of NPs to polymers on the assembly morphologies. The present work expands on the mechanism of polymer-guided assembly of inorganic NPs in films.

The presented work is sound, and experimental data gives deep insight into the assembly of polymer-NP composites. There is no doubt that this paper reports significant advancement in a field. Thus, I would recommend the publication of this work in Nature Communications, after few concerns are addressed:

1. The authors carefully characterized the interactions between NPs and PBLG. However, it is still not straightforward for readers to understand how the lamellar structures are formed on the basis of these interactions. I would suggest the authors include a new scheme illustrating the mechanism of microphase separation in the main text or SI.

2. Is it possible that the immiscibility between the exposed surface of NPs and homopolymers is the driving force of phase separation? One quick experiment may be considered to verify this possibility: mixing and assembling NPs with homopolymers (e.g., PS) with the same end functional group.

3. What is the thickness of the films? Is the assembly morphology dependent on the film thickness?

4. The formation of Moiré patterns in the assembly is interesting and rarely observed in polymer systems. Why? A reasonable explanation would be very helpful.

5. Some figure captions are not clear:

a. "Figure 1 Building blocks...", but this figure includes both building blocks and assembly structures.

b. "...Evolution of characteristic widths as a function of the degree of polymerization x : squares:

average periodicity of the lamellas; white circles: average width of the white...". The format is a bit confusing.

Response to the reviewers.

Reviewers' comments:

Reviewer #1 (Remarks to the Author):

The authors produced mixtures of nanoparticles and a homopolymer in THF as solvent. The nanoparticles were very spherical Pt nanoparticles with a diameter of 1.2 ± 0.3 nm as determined with electron microscopy. The homopolymer was poly(γ -benzyl-L-glutamate) with different molar masses. In the absence of nanoparticles this polymer forms an alpha helix in THF.

The new finding of the work is that lamellar nanostructures are formed upon mixing. This lamellar phase consists of alternating lamellae of type 1 and type 2. The lamellae of type 1 are enriched in nanoparticles while no nanoparticles are present in type 2. This was visualized with TEM and found for a wide range of compositions.

Similar nanostructures are well reported for block-copolymers but not for homopolymers. The explanation for such nanoparticle-enriched lamellae in block-copolymers is straight-forward as can be found in the literature cited by the authors.

Normally one expects that either the polymer decorates the nanoparticle or vice versa the nanoparticles decorate the polymer. Such depend to the molar masses, concentrations, strength of interaction of the polymers and the nanoparticles etc.

But the formation of lamellar structures of the named homopolymer is highly unexpected and to me not sufficiently explained by the authors.

Following the remarks of Reviewer #1, and suggestions of Reviewer #2 and Reviewer #3, several additions has been made in the manuscript in order to explain better the formation of the lamellar structure.

A paragraph has been added to describe the self-assemblies at short times of mixing (p. 7), the Discussion section is now more detailed (p. 7-8) and an explicative figure has been added in SI (Fig. S18).

For each polymer chain the nanoparticles need to induce a change of the conformation from an alpha helix to a random coil. But only for that fraction of the chain which is then located in the nanoparticle-rich lamellae type 1. The remaining part of each chain is still in an alpha-helical conformation forming lamellae type 2. When assuming this mechanism, a kind of pseudo block-copolymer is formed: therein block 1 is the random coil and block 2 is the alpha helix. In principle such is possible and can explain the lamellar structures. But such scenario is very unlikely. Instead, I assume that the lamellar structures are formed during deposition of the mixtures on surfaces prior to TEM measurements. It needs to be excluded that a phenomenon similarly to the coffee ring effect produces the lamellar structures. Further experiments for ensemble averaging data are needed.

The coffee ring effect is a pattern left by a puddle of particle-laden liquid after it evaporates. It originates from combination of evaporation of solvent, capillary flow and Marangoni flow. It has a purely physical origin. Such an effect has been observed in our systems, and influenced the morphology of the assemblies at the micrometer scale, as shown in the picture below.

We prefer avoiding speaking about coffee ring effect in the present manuscript, as it is in the margin of the main message: a description of the superstructures at the nanometer scale, where the main driving forces of the assembly are dipolar interactions and molecular interaction, such as Van der Waals forces or coordination bonds.

The authors are already going in the right direction with their SAXS measurements. But the SAXS data presented in Figure S6 is absolutely not sufficient. The peak in the scattering curve could simply result from a so-called correlation hole as found for many polymeric structures. The peak is no proper evidence for the existence of a lamellar structure in solution. Here a reasonable modelling of the SAXS curves is needed and data should be measured from all the samples that correspond to the TEM pictures presented in Figures 1 and 2. The SAXS measurements should be performed from samples in solution in a capillary. Otherwise the data are prone to artifacts. A normal lab SAXS instruments is suitable. The lamellar structures must provide a very strong scattering intensity and a distinctive scattering pattern, if they exist in solution as claimed by the authors. Data evaluation can be performed with open source small-angle scattering programs. These programs provide appropriate models for lamellar structures to simulate the measured curves.

As asked by Reviewer #1, SAXS measurements have been performed on assemblies with **PBLG1 to PBLG5**. In addition to confirming a constant inter-particle average distance, such measurements proved the coherence of the lamellar structure at the macroscopic scale and confirmed an increase of the inter-lamellar distance with the degree of polymerization of the PBLG.

The corresponding figure and a table comparing the periodicities in TEM and SAXS have been added in SI (Fig. S14 and Table S2). A new sentence has been added in the main manuscript (p. 6), and the experimental section has been completed accordingly (p. 11). Pierre Roblin, who conceived and performed the SAXS measurements, has been added to the list of co-authors.

In summary, the manuscript needs major revision that provides clear evidence for the presence of the lamellar structure in solution.

To get signals in SAXS, we had to concentrate the samples and measurements were performed at the beginning of precipitation. To provide clear evidence of the presence of the lamellar structure in solution, we also performed cryo-TEM microscopy (as suggested by Reviewer #2). After adaptation of the classical cryo-TEM techniques to work with THF, we succeeded in observing the lamellae in solution.

This supplementary characterization provides a direct and indisputable proof of the presence of lamellae in solution. A corresponding figure has been added in SI (Fig. S4) and a descriptive sentence has been added in the main text (p. 4). Stéphanie Balor, who conceived and performed the cryo-TEM imaging, has been added to the list of co-authors.

Reviewer #2 (Remarks to the Author):

The entitled work “Beyond Flory-Huggins theory: bidimensional lamellar assembly by coordination of peptidic homopolymers to platinum nanoparticles” by Manai et al. reported the use of homopolymers to control the organization of nanoparticles. The authors confirmed that PBLG polypeptide homopolymers coordinated with nanoparticles by ¹³C NMR spectroscopy. We could imagine that the homopolymers formed brush layers on the surfaces of nanoparticles, which makes this system perfectly fit into the frame of the Flory-Huggins theory. In previous studies, the phase behaviors of the conjugates (or amphiphiles) of nanoparticles, e.g. Au NPs (Mei et al. Chem Eur J 2019, 1) or POSS (Wu et al. Macromolecules 2014, 4622; Zhang et al. Soft Matter 2019, 7108), and polymers have been well documented. The PBLG-Pt nanoparticle conjugates in this work were in-situ generated upon the mixing of PBLG and Pt nanoparticles, which made it different from previous reports on ex-situ synthesized nanoparticle-polymer conjugates. This work does not have enough novelty for publication in Nature Communications.

Whereas the references proposed by Reviewer #2 are very interesting, the mechanism proposed therein cannot describe the system presented in our manuscript. First, we observed coordination of the peptide bonds of the polymers to the nanoparticles surface. This experimental observation clearly rules out the formation of simple conjugates composed of two "blocks", *i.e.* in our case, nanoparticles only linked by the end-alkyne terminal groups of the polymer. Second, in the references mentioned by Reviewer #2, the rod behavior of the system is provided by the inorganic part of the conjugate; we then agree that the behavior of the organic part is in agreement with the Flory-Huggins theory. In marked contrast, it is the organic part – the PBLG part adopting α -helical conformation – that exhibits the rod-like behavior in our system. Based on the data from Figure 4 and Table S2, we can establish a scaling law of the lamellae thickness *vs.* the average molar mass proportional to $M_n^{0.98}$ (see the figure below). With diblock copolymers, a bibliographic study indicates that a scaling law exponent of lamellae thickness *vs.* molar mass varies experimentally from 0.5 for the lower masses (*e.g.* < 10000 g/mol) to 0.7 for the larger ones (Sandre Olivier et al., Polymer, 2010, 21 (21), 4673-4685). These values are in principle close to the reference exponent of 0.643 predicted by Helfand (Helfand, E., Macromolecules 1975,8, 552-556; Helfand, E.; Wasserman, Z. R., Macromolecules 1976,9, 879-888), which is significantly lower than the exponent value of 0.98 estimated from our data.

Additional comments:

1. The authors claimed that the lamellar structures formed in the solution, but they characterized the films after the evaporation of solvent. It would be more convincing to use cryo-TEM to characterize the solution structure.

The experiment has been done and gave convincing results. We thank Reviewer #2 for his suggestion.

A corresponding figure has been added in SI (Fig. S4) and a descriptive sentence has been added in the main text (p. 4).

2. The exact structure is still unclear. It will be helpful to capture the structures at the early stage and characterize the kinetics of the self-assembly process.

Regular experiments are done with 2 hours of agitation to be sure to reach a steady state. Following Reviewer #2's suggestion, we explored very short time of reactions and noticed the structures are generally formed after 5 min. Specific experiments have thus been done to catch the very beginning of the self-assembly process, which turns out to start in the first seconds of mixing.

To take into account this comment, a descriptive paragraph has been added in the main text (p. 7), and a corresponding figure has been added in SI (Fig. S17).

3. The use of "Beyond the Flory-Huggins theory" in the title might not be appropriate.

Even if we are convinced that our systems cannot be described by the conventional formalism developed for diblock-copolymer, as detailed above, we decided to modify the title in line with Reviewer #2's comment.

Reviewer #3 (Remarks to the Author):

The manuscript by Manai et al. describes the self-assembly of a mixture of small platinum nanoparticles (NP) and poly(benzyl-L-glutamate) (PBLG) into periodic lamellar structures. The formation of microphase-separated structures is driven by the coordination interactions between the surface of NPs and the functional groups of PBLG. The authors systematically investigated the effect of substrates, polymer length, and volume ratio of NPs to polymers on the assembly morphologies. The present work expands on the mechanism of polymer-guided assembly of inorganic NPs in films.

The presented work is sound, and experimental data gives deep insight into the assembly of polymer-NP composites. There is no doubt that this paper reports significant advancement in a field. Thus, I would recommend the publication of this work in Nature Communications, after few concerns are addressed:

1. The authors carefully characterized the interactions between NPs and PBLG. However, it is still not straightforward for readers to understand how the lamellar structures are formed on the basis of these interactions. I would suggest the authors include a new scheme illustrating the mechanism of microphase separation in the main text or SI.

A corresponding figure has been added in SI (Fig. S18) and the Discussion section of the main text is now more detailed (p. 7-8).

2. Is it possible that the immiscibility between the exposed surface of NPs and homopolymers is the driving force of phase separation? One quick experiment may be considered to verify this possibility: mixing and assembling NPs with homopolymers (e.g., PS) with the same end functional group.

The experiment has been done using PS with a DP of 336, in the same order of magnitude as the one used for PBLG in exactly the same experimental conditions. We see in TEM imaging (see picture below - left) that some kind of structuration appears, but clearly different from the well-ordered lamellae obtained with PBLG. Immiscibility between the homopolymer and the NP zone thus clearly plays a role in the structuration, but the presence of coordinating groups is crucial to go a step further and interact with the NP, as detailed throughout the manuscript. If the system is only governed by repulsive forces, arrangement between the components generally tends to form isotropic structure (like the spherical aggregates observed with PS – see picture below - right). There might be some interest to study such a phenomenon in more details in the scope of a separate study.

3. What is the thickness of the films? Is the assembly morphology dependent on the film thickness?

Analysis of the tomography picture already obtained for **PBLG4** showed that the thickness of the film is comparable to the width of the NP lamella. We performed tomography microscopy with **PBLG3** to confirm this trend.

A corresponding figure has been added in SI (Fig. S15) and a descriptive sentence has been added in the main text (p. 6).

4. The formation of Moiré patterns in the assembly is interesting and rarely observed in polymer systems. Why? A reasonable explanation would be very helpful.

Thanks to the present revision of the manuscript, we have demonstrated that the assembly formed in solution. If the lamellae are pre-formed in solution, they can deposit on top of each other during the drop casting of the TEM grid preparation, if the solution is sufficiently concentrated. We should notice that such an effect has been observed by chance. It will be of interest to study it in more details the scope of another work. But we think it is interesting to mention it in the present manuscript as it gives a supplementary indirect proof of the existence of the lamellae in solution.

A sentence of explanation as been added in the main text (p. 4).

5. Some figure captions are not clear:

a. "Figure 1 Building blocks...", but this figure includes both building blocks and assembly structures.

The caption has been corrected (p. 16).

b. "...Evolution of characteristic widths as a function of the degree of polymerization x: squares: average periodicity of the lamellas; white circles: average width of the white...". The format is a bit confusing.

A legend has been added in the figure and the caption has been adapted (p.19).

REVIEWERS' COMMENTS:

Reviewer #2 (Remarks to the Author):

I agreed with the argument in the response "This experimental observation clearly rules out the formation of simple conjugates composed of two "blocks", i.e. in our case, nanoparticles only linked by the end-alkyne terminal groups of the polymer. " It did not form a `diblock" conjugate, but they are hairy nanoparticles with PLBG as the corona. The difference is that PLBG had rod-like conformation, while the polymers in previous studies might form random coils. I suggested the authors acknowledging previous works on nanoparticle-polymer conjugates in the revised version.

Reviewer #3 (Remarks to the Author):

The authors have performed additional experiments and addressed the comments raised by the reviewers. The revised manuscript is now suitable for publication in Nature Communications as is.

Response to the reviewers.

Reviewers' comments:

Reviewer #2 (Remarks to the Author):

I agreed with the argument in the response "This experimental observation clearly rules out the formation of simple conjugates composed of two "blocks", i.e. in our case, nanoparticles only linked by the end-alkyne terminal groups of the polymer. " It did not form a 'diblock' conjugate, but they are hairy nanoparticles with PLBG as the corona. The difference is that PLBG had rod-like conformation, while the polymers in previous studies might form random coils. I suggested the authors acknowledging previous works on nanoparticle-polymer conjugates in the revised version.

The review suggested by Reviewer #2 in the previous comments has been added [Mei, S., Staub, M. & Li, C. Y. Directed Nanoparticle Assembly through Polymer Crystallization. *Chem. – Eur. J.* **26**, 349–361 (2020).], as well as another recent one [Yi, C., Yang, Y., Liu, B., He, J. & Nie, Z. Polymer-guided assembly of inorganic nanoparticles. *Chem. Soc. Rev.* **49**, 465–508 (2020).]. They are now refs 2 and 3 in the new version of the manuscript.

Reviewer #3 (Remarks to the Author):

The authors have performed additional experiments and addressed the comments raised by the reviewers. The revised manuscript is now suitable for publication in Nature Communications as is.